# Trauma Characteristics Associated with E-Scooter Accidents in Switzerland—A Case Series Study

**DOI:** 10.3390/ijerph20054233

**Published:** 2023-02-27

**Authors:** Ava Insa Bracher, Samuel Klingler, Sabine Koba, Dominik Andreas Jakob, Aristomenis Exadaktylos, Jolanta Klukowska-Rötzler, John-Patrik Burkhard

**Affiliations:** 1Department of Emergency Medicine, Inselspital, University Hospital Bern, University of Bern, 3010 Bern, Switzerland; 2Department of Oral Surgery and Stomatology, School of Dental Medicine, University of Bern, 3010 Bern, Switzerland; 3Cleft and Craniofacial Center Professor J. A. Obwegeser, 8005 Zurich, Switzerland

**Keywords:** electric scooter, e-scooter, e-mobility, traffic accident, facial fracture, craniocerebral trauma

## Abstract

E-scooters have gained popularity worldwide in the last few years. Due to the increase in users, more accidents related to e-scooters can be observed. The present study aimed to analyse epidemiological data, characteristics, and severity of injuries in patients admitted to a Level I trauma centre in Switzerland (Inselspital Bern, University Hospital Bern) after accidents associated with e-scooters. This retrospective case series evaluated 23 patients who presented to the University Hospital of Bern between 1 of May 2019 and 31 of October 2021 after an e-scooter accident. Data were collected on patient demographics, time and cause of the accident, speed, alcohol consumption, helmet use, type and localisation of injury, number of injuries per patient, and outcome. Men were most frequently affected (61.9%). The mean age was 35.8 (STD 14.8) years. Slightly more than half (52.2%) of all accidents were self-inflicted. Most accidents were reported during the night (7 p.m. to 7 a.m., 60.9%) and in summer (43.5%). Alcohol consumption was reported in 43.5% of cases, with a mean blood alcohol level of 1.4 g/l. Most injuries were observed in the face (25.3%) and head/neck area (20.25%). Skin abrasions (56.5%) and traumatic brain injury (43.5%) were the most common types of traumata in terms of total number of patients. Only in one case it was reported that a protective helmet had been worn. Five patients required hospitalisation and four patients underwent surgery. Three patients underwent emergency orthopaedic surgery, and one patient underwent emergency neurosurgery. E-scooter accidents result in a significant number of facial and head/neck injuries. E-scooter riders would potentially benefit from a helmet to protect them in the event of an accident. Additionally, the results of this study indicate that a significant number of e-scooter accidents in Switzerland occurred under the influence of alcohol. Prevention campaigns to raise awareness of the risks of driving e-scooters under the influence of alcohol could help prevent future accidents.

## 1. Introduction

Micromobility is a rapidly growing trend, defined as a range of small, lightweight vehicles, either human- or electrically powered, that typically travel at low (25 km/h top speed) or moderate speeds (45 km/h top speed), such as scooters, bicycles, Segways, and skateboards. Electric micromobility devices, including e-bikes and e-scooters, are attractive to nearly everyone as they allow travel of an increased area at a higher speed. Additionally, they are worthwhile alternatives to short-trip car journeys and valuable extensions to public transport. In cities, where many inhabitants do not own cars, e-bikes and e-scooters are getting more and more popular [1]. Moreover, the COVID-19 pandemic induced a desire in the population to commute outside of crowded buses and subways, thus rendering a private or shared micromobility device even more attractive [2].

Stand-up electric scooters (e-scooters) have gained popularity worldwide in the last few years. Currently, no data are available for private e-scooter sales in Switzerland. Given that the number of e-bikes sold in Switzerland quadrupled from 2010 to 2020 (from 40,000 to 170,000 pieces p.a), it is very likely that the number of e-scooters sold also increased accordingly [3]. Due to the ease of sharing, e-scooters have become the preferred vehicle for short-distance travel, especially in big cities [2,4,5,6]. In Bern, the capital of Switzerland, two e-scooter sharing companies, Voi and Tier, were introduced in 2021, thus matching the worldwide trend. 

Due to the increase in e-scooter users, more accidents related to e-scooters can be observed [2,7,8,9,10,11]. In most cases, the associated injuries can be treated quickly and conservatively. However, severe injuries, especially head injuries and fractures, require intensive care monitoring, surgical intervention, and hospitalisation. Aside from the impact on the personal health of the injured individual, the resulting diagnostics, hospital stays, and illness-related absences have socio-economic consequences. For the reasons mentioned above, accidents involving e-scooters are an important topic of current relevance for emergency departments (ED), although available data are still scarce.

The present study aimed to analyse epidemiological data, characteristics, and severity of injuries in patients admitted to a Level I trauma centre in Switzerland (Inselspital Bern, University Hospital Bern) after accidents associated with e-scooters.

## 2. Materials and Methods

### 2.1. Study Design and Sample

This single-centre retrospective case series study was based on the demographic- and health-related data of all patients being admitted to or were presented at the Level I interdisciplinary University Emergency Department (ED) at the Inselspital Bern after an e-scooter accident during the selected 30-month period from 1 May 2019 to 31 October 2021. Our ED is responsible for the emergency treatment of approximately 51,815 (2021) patients per year from a catchment area of 2 million people. All patients were adults (≥16 years old) besides 1 13-year-old and 1 15-year-old. Generally, only patients older than 16 years are treated at the ED, as younger patients are referred to the children’s hospital. However, due to extended diagnostic and interventional facilities, severe trauma cases in patients younger than 16 years of age are assigned to the ED. Patients with injuries related to non-electrical scooters were excluded. Moreover, patients without general consent (GC), as well patients who actively refused GC, were excluded from the study. The patient records of the University ED were searched using the search term “e-scooter”. A total of 24 patients were admitted to the emergency department for an e-scooter accident during the study period. One case was mistakenly recorded as an e-scooter accident but was an accident with a knee scooter; therefore, this patient was excluded from the study.

### 2.2. Data Collection and Extraction

All data were extracted from medical records and files stored in the clinical database system (E.care ED, ED 2.1.3.0, E.care BVBA, Turnhout, Belgium). Relevant information included age, gender, time of accident (daytime [from 7 a.m. until 7 p.m.] vs. nighttime [7 p.m. until 7 a.m.]), season (spring, summer, autumn, winter [defined according to the astronomical timing of the northern hemisphere]), cause of accident (external impact vs. no external impact vs. unknown circumstances), velocity (estimations of patients and witnesses), influence of alcohol (g/L) or drugs at the time of accident, wearing of helmet or other protective gear, as well as the main type of injury, the number of injuries per patient, and the outcome (home vs. hospital vs. surgery). 

### 2.3. Statistical Analysis

Descriptive statistics were utilized to determine the frequencies and percentages for dichotomous variables, the mean and median values, standard deviation, and ranges of numerical variables. The distribution of continuous variables was described as the mean and standard deviation for normally distributed variables. Median and interquartile range was used to report normally distributed variables. The distribution of categorical data was reported as numbers and percentages. The interobserver reliability was assessed with Cohen’s kappa statistic. Diverging values were discussed and a consensus was found. Data analysis was performed using the Statistical Package for Social Sciences for Macintosh (SPSS^®^ Version 24.0, IBM Corp., Armonk, NY, USA).

### 2.4. Ethical Considerations

The Cantonal Ethics Committee of Bern approved this study (KEK-BE 2021-02132) which followed the guidelines of the Declaration of Helsinki and ethical principles for conducting medical research with human subjects [12]. Data were processed according to the standards of the Ethics Committee and Swiss law. Following the recommendation of the ethics committee, all patients were given information about the study and had to agree to the inclusion of their data in the research. The analysis was carried out with anonymized data.

### 2.5. Patient and Public Involvement

Patients and the public were not involved as co producers of this research.

## 3. Results

A total of 23 (*n* = 23) accidents with e-scooters were reported during the 30-month study period from May 2019 until October 2021. All patients agreed to the inclusion of their anonymized data in the research. Less than half of all patients (39.1%, *n* = 9) were female and 60.9% male (*n* = 14). Patient ages ranged from 13 to 63 years, with a mean age of 36 (STD 14.8) years. Detailed patient characteristics are shown in Table 1.

Of all accidents, 60.9% (*n* = 14) happened during nighttime, whereas only 39.1% (*n* = 9) of e-scooter drivers had their accident during the daytime. Most of the accidents occurred in summer (43.5%, *n* = 10) and spring (34.8%, *n* = 8); during autumn months four cases (17.4%) were registered and only one (4.3%) during wintertime. The circumstances were known for most of the accidents: in 52.2% (*n* = 12), no external influence was involved, whereas in 26.1% (*n* = 6) another human influenced or caused the evolvement of the accident. Four patients had direct collisions with cars, one of them with a parked car. The remaining two patients were injured during evasive maneuvers. In five cases (21.7%), the history of the accident was unknown or could not be determined from the data available. Four patients had their accident under the influence of alcohol.

In total, 43.5% (*n* = 10) of the accidents happened with inebriated e-scooter drivers; 30% (*n* = 3) were female, 70% (*n* = 7) were male. The proportion of patients who were intoxicated were 7 of 14 men (50%) and 3 of 9 (33%) women, (*p* = 0.669). During nighttime, the majority of all accidents occurred under the influence of alcohol [8 out of 14 accidents (57%)] and without any external influence [7 out of 14 accidents (50%)]. Data regarding blood alcohol amount were recorded for seven patients only with a median level of 1.4 g/L (lower quartile 1.2 g/L, upper quartile 1.6 g/L). Drug abuse was recorded in four cases with additional alcohol consumption in 75% (*n* = 3).

The speed at the time of accident was recorded in 10 cases with a median of 20 km/h (lower quartile 15.6 km/h, upper quartile 20 km/h). A helmet was worn by one patient only (4.3%). For 69.6% (*n* = 16) there were no data available. In 26.1% (*n* = 6) of cases, medical records indicated that a helmet was not worn at the time of the accident.

Table 2 summarises the locations and types of injuries. The percentage values are in relation to the total patient amount, *n* = 23. The most frequent types of injuries were dermal abrasions, seen in 56.5% of all patients (*n* = 13), craniocerebral traumas (CCTs) in 43.5% (*n* = 10), and facial contused lacerated wounds (CLWs) in 39.1% (*n* = 9). Of all e-scooter accident patients examined, 30.4% (*n* = 7) had a fracture of their upper extremity. Facial fractures and fractures of the lower extremities occurred in 26.1% (*n* = 6), respectively.

A total of 79 injuries were found in all 23 patients, and a single patient could have more than one injury. The number of injuries per patient varied from 1 injury up to 11 injuries, with a median value of 3 injuries per patient (lower quartile 2 injuries, upper quartile 4 injuries). Most of the injuries occurred in the facial area (25.32%), the head and neck region (20.25%), and the skin (16.46%), as shown in Figure 1. Of all injuries counted, only 15.19% were assigned to the upper extremities and 12.66% to the lower extremities. The chest (6.33%) and the abdomen (3.80%) were less frequently harmed.

Most patients (78.3%, *n* = 18) were discharged home or for further outpatient treatment after diagnostics and treatment in the ED. Only five patients (21.7%) required hospitalisation; 80% (*n* = 4) of them underwent surgery, including two patients who had to undergo an emergency craniectomy. None of the e-scooter drivers died during the time examined in this study.

## 4. Discussion

Since their introduction in 2017 in San Francisco, USA, e-scooters have quickly swept the world. As a result, medical studies from almost every continent are now evaluating e-scooter accidents. To our knowledge, this is the first study to investigate e-scooter accidents in Switzerland. Our aim was to provide a preliminary assessment of epidemiological data and characteristics of injuries related to e-scooter accidents over a 30-month period and to compare the results with already available data from other countries.

E-scooter accidents frequently involve young adults, with males outnumbering females in most studies [7,9,13,14,15,16,17,18]. Only Blomberg et al. (2019) described a different gender distribution, with 57.1% of the injured e-scooter pilots being female [5]. Possible reasons for a higher incidence of e-scooter accidents among young male adults could be a more risk-taking attitude in general, and higher alcohol consumption compared to women. The present study supports this assumption, as 70% of all alcohol-related accidents involved male patients. In Switzerland, the legal limit for alcohol in road traffic is 0.5‰. Considering that the median level of blood alcohol level in the patients assessed was 1.4 g/L (corresponding to approximately 1.4‰), it is not surprising that most of the accidents were self-inflicted (52.2%) or the cause of the accidents were unknown (21.7%). Clearly, intoxication plays a significant role in e-scooter accidents; Gan-El et al. (2022) reported intoxication in 30% of their patients, Blomberg et al. (2019) in 36.6%, Mair et al. (2021) in 36.7%, and Grill et al. (2022) in 52.5% of the e-scooter users admitted to hospital [5,16,17,18]. This is similar to our findings of 43.5%. Oksanen et al. (2020) even found as many as 91% of all patients intoxicated, with an average blood alcohol level of 1.7‰ [13]. In September 2020, the rental companies Voi introduced an in-app reaction test to investigate users’ fitness to drive [19]. However, riders are still able to unlock the scooter regardless of whether or not they pass this test. A significant number of e-scooter accidents in Switzerland occurred under the influence of alcohol. Alcohol use is of concern because it impairs a driver’s judgment and may lead to accidents. Prevention campaigns raising awareness of the risks of driving e-scooters under the influence of alcohol should be specifically promoted. In addition, more rigorous alcohol controls and consecutive punishment of e-scooter drivers under the influence of alcohol could help prevent future accidents.

Considering the time of day, authors from other studies describe too an increased number of e-scooter accidents during evening and nighttime hours, which is in line with our findings [5,8,9,16,17,18]. Trivedi et al. (2019) reported 75% of all e-scooter-related admissions occurred between 3 p.m. and 7 a.m. [8]. Blomberg et al. (2019) observed even more admissions (83%) for the same period [5]. Both time windows are larger than the one in our study (7 p.m.–7 a.m.). However, Moftakhar et al. (2021) defined nighttime similarly to us from 8 p.m.–8 a.m. and reported that 58.3% of all accidents occurred during this period [9], which is strongly comparable to our result of 60.9%. Aside from higher consumption of alcohol and drugs during evening hours and nighttime, poorer visibility when driving, fatigue, and lack of concentration are also possible factors contributing to more accidents in the evening hours. Additionally, most e-scooters are very quiet, thus other road users might struggle to hear them in the dark. Since most accidents occur at night, improved lighting for e-scooters is worth discussing. Other authors even suggested a night driving ban for e-scooters to reduce the number of accidents. However, it seems very unlikely that a night ban can be implemented in Switzerland.

From a seasonal point of view, almost half of all accidents occurred in the summer months (43.5%), which we defined according to the astronomical timing of the northern hemisphere from the end of June to the end of September. It is beneficial to compare these findings with results from studies in a similar geographic setting, which has roughly the same distribution of climate and seasons. Two studies from Germany confirm our findings, reporting most of the accidents from July to August for Cologne [4], and the peak of accidents from August to September in Frankfurt [20]. This supports the assumption that in Central Europe, where the seasons are accompanied by different climate conditions, e-scooter use is generally preferred during warm and dry weather conditions, leading to more accidents during this period. However, it needs to be taken into account that the seasons are not evenly distributed across our observation period.

Regarding the localisation of injuries in terms of anatomical areas, in our study most were observed in the face and head/neck area (45.55%). Even in international comparison, head and face injuries seem to be the most frequent, closely followed by upper and lower limb injuries [5,8,9,11,16,17,21]. Bekhit et al. (2020) found most injuries in the upper limb (37.1%) and only 20.4% in the head and neck region [22]. Gan-El et al. (2022) and Moftakhar et al. (2021) both reported most injuries in the upper limbs, closely followed by the head and neck area [9,18]. The frequent occurrences of head and neck injuries in conjunction with injuries to the upper extremities might indicate that drivers attempted to break their fall before hitting the ground. In comparison, Oksanen et al. (2020) recorded only craniofacial injuries, probably due to the high number of intoxicated patients (91%) and the consequent inability to break the fall with their hands [13]. To verify the hypothesis that intoxication may lead to a higher number of head and neck injuries, further studies with a structured analysis of substances consumed and blood levels are needed.

In the present study, skin abrasions were observed in 56.5% of all patients. When comparing our data to the literature available, we realised that many authors did not assess skin abrasions as a separate category but listed them under the respective anatomic region. However, Moftakhar et al. (2021) also categorised abrasions separately, along with lacerations and hematomas. In line with our findings, they described that this category of soft tissue injury was the most common injury overall [9].

Next to skin abrasions, craniocerebral traumas (CCTs) were the second most common type of trauma, accounting for 43.5% in this study. These were mainly Grade I traumatic brain injuries that required merely neurological monitoring, allowing most patients to be discharged home. Only in two cases did a severe cerebral haemorrhage lead to emergency surgery. These cases involved a collision with an oncoming car, which explains the extent of the injuries by the severity of the impact. However, most patients had rather minor head injuries, although they were probably not wearing helmets and often intoxicated.

Nevertheless, wearing helmets significantly reduces the likelihood of severe head injuries in cyclists involved in accidents [23,24]. Amongst e-scooter drivers, helmet use is generally not widespread, as confirmed by many available studies [2,4,7,8,9,10,11,16,17,18,20,25]. One of the first studies regarding e-scooter injuries was conducted by Trivedi et al. (2019) from September 2017 to August 2018. Although California law required the wearing of a helmet when riding an e-scooter throughout the whole study period, only 4.4% of all patients complied with the law [8], thus questioning the use of a legal obligation of helmet protection. Interestingly, this is in line with our results; however, there is no helmet obligation for e-scooters in Switzerland. On the other hand, Mitchell et al. (2019) describe a considerably positive effect of mandatory helmet use. During their study period, a compulsory helmet law was installed leading to a compliance in helmet use in almost 50% of all patients. Furthermore, patients who wore a helmet were less likely to have a severe head injury compared to those who did not [21]. This was confirmed by Hamzani et al. (2021), who found that wearing a helmet could reduce the incidence of fractures as well as dentoalveolar injuries [26]. However, they reported an increased occurrence of soft-tissue injuries in patients wearing a helmet. This was linked to a potential breakage of the helmet’s plastic visor on impact [26]. Two recent studies evaluated the use of protective helmets in prevention of severe head injuries in e-scooter accidents, using simulations with human body models [27,28]. The results suggest that helmets have the potential to reduce the impact forces experienced by the head during a fall, lowering the risk of head injuries. It is important to note that further research is necessary to determine the optimal design and effectiveness of helmets in reducing head injury risk in e-scooter accidents. However, based on the current evidence, both models suggest that wearing a helmet while riding an e-scooter is a simple and effective way to reduce the risk of head injuries in the event of a fall. Each country must consider whether its population would benefit from an official helmet obligation, with possible fines for violations. Based on the available data, however, there seems to be a general benefit in helmet protection. The rental company Tier offers foldable helmets for free [29]. On their “How to use” webpage, however, they do not include any advice or information about the usage of the integrated helmets. Only in a YouTube video is this step further explained. In our opinion, to increase the usage of helmets by e-scooter drivers, the availability of foldable helmets should be promoted more strongly.

Currently, e-scooter drivers in Switzerland must use public roads along with other traffic. Some of them drive illegally on pavements to avoid cars and buses. Separate lanes for cyclists and e-scooter users are still scarce in Swiss cities. Separate lanes are desirable as they could improve safety for both micromobility users and pedestrians. However, it must be taken into account that most e-scooter incidents occur due to the inattentiveness or risky driving of the e-scooter users themselves and not through the influence of others [10]. This is confirmed by our data, which show that in only 26.1% of cases another human influenced or caused the accident.

Several limitations were observed in this case series. The catchment area of the investigated Level I trauma centre is limited compared to other, much bigger European cities, although the emergency department of the Inselspital Bern covers a large part of Switzerland. In addition, the database studied includes only patients who presented to the ED. This could lead to under-representation, as many patients suffering from milder injuries are likely to be treated by their family doctors, dentists, or smaller EDs in the surrounding area [22]. Due to the retrospective nature of the study, details of interest, such as helmet protection and blood alcohol levels, were not assessed in a standardised manner. A prospective study over a longer period involving multiple trauma centres in Switzerland could provide more information on different regions and lead to a larger patient population. However, considering the international data, a similar picture as in the present study is likely to emerge.

## 5. Conclusions

E-scooter accidents lead to a considerable amount of face and head/neck injuries. As literature available underlines the positive effects of helmet protection in cyclists as well as e-scooter drivers, it is beneficial for both groups to use them. In addition, the results of this study indicate that a significant number of e-scooter accidents in Switzerland occurred under the influence of alcohol. Prevention campaigns to raise awareness of the risks of driving e-scooters under the influence of alcohol could help prevent future accidents.

## Figures and Tables

**Figure 1 ijerph-20-04233-f001:**
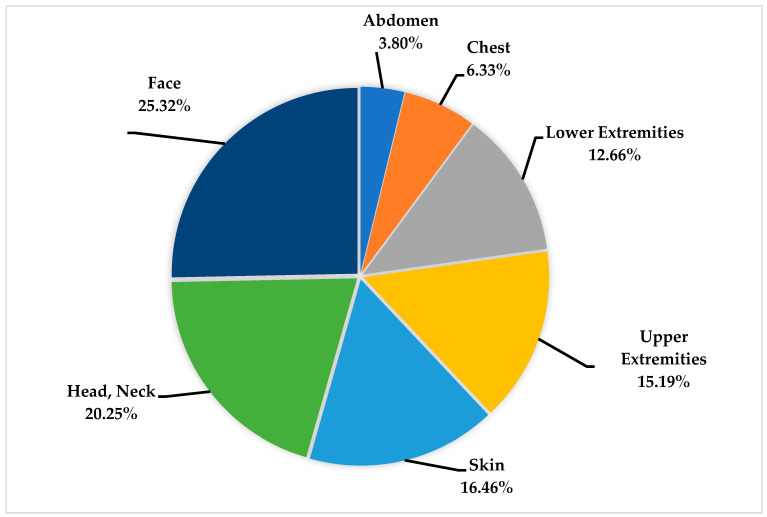
Percentage of injured regions in relation to a total of 79 injuries counted.

**Table 1 ijerph-20-04233-t001:** Data on e-scooter drivers.

n	Sex	Age	Time	Season	Cause of Accident	Velocity (km/h)	Alcohol	Alcohol Level in g/L	Drugs	Helmet/ Protective Gear	Type of Principal Injury	Region of Main Injury	Number of Injuries	Outcome
1	f	32	night	spring	external impact	12.5	no		no	none	CLW	chin	4	discharge
2	m	37	day	spring	no external impact	15	no		no	*	pain	lower back	1	discharge
3	f	38	night	autumn	no external impact	17.5	no		no	*	CCT	head	2	discharge
4	f	37	night	autumn	no external impact	*	no		no	*	CLW	face	2	discharge
5	f	33	night	winter	*	40	yes	*	no	none	CCT	head	2	discharge
6	m	47	night	spring	*	30	yes	1.4	no	none	CCT	head	4	discharge
7	f	57	day	spring	no external impact	20	yes	1.8	no	*	fractures	face	3	discharge
8	m	16	day	summer	external impact	*	no		no	none	CCT	head	1	discharge
9	m	26	night	summer	no external impact	20	yes	1.2	no	*	fractures	teeth	4	discharge
10	m	57	day	summer	no external impact	*	no		yes	*	fractures	Upper + lower extremity	3	Orthopaedic surgery (ORIF tibia and radius)
11	m	31	day	summer	external impact	*	no		no	*	fracture	foot	2	discharge
12	m	43	night	summer	no external impact	20	yes	1.2	yes	*	CCT	head	3	discharge
13	m	27	night	autumn	*	*	yes	1.4	yes	*	fractures	face	2	discharge
14	f	20	day	autumn	external impact	*	no		no	*	contusion	upper + lower extremity	3	discharge
15	f	24	night	spring	*	*	yes	1.9	no	*	CCT	head	4	discharge
16	m	40	day	spring	no external impact	15	yes	0.9	no	*	fracture	lower extremity	2	Orthopaedic surgery (ORIF tibia)
17	m	20	night	spring	no external impact	*	yes	*	no	*	contusion	upper extremity	1	discharge
18	m	43	night	spring	no external impact	*	no		no	none	fracture	upper extremity	2	discharge
19	m	61	day	summer	no external impact	20	no		no	*	CCT	head	4	discharge
20	m	44	night	summer	no external impact	*	yes	*	yes	none	CLW	face	4	discharge
21	m	63	day	summer	*	*	no		no	yes	ICH	head	6	hospital
22	f	13	night	summer	external impact	*	no		no	*	ICH	head	9	Orthopaedic surgery (TEN osteosynthesis humerus fracture), neurosurgery (craniectomy), maxillofacial surgery (ORIF mandibular fracture)
23	f	15	night	summer	external impact	*	no		no	*	ICH	head	11	neurosurgery (craniectomy), orthopaedic surgery (CRIF tibia)

Abbreviations: CLW = contused lacerated wound, CCT = craniocerebral trauma, ICH = intracranial haemorrhage, * = missing data.

**Table 2 ijerph-20-04233-t002:** Number and type of injuries for each body region in relation to total patient amount (*n* = 23).

Head and Neck	CCT *n* = 10 (43.5%)	ICH *n =* 3 (13%)	Skull fracture *n =* 2 (8.7%)	CLW *n =* 1 (4.3%)
Face	CLW *n* = 9 (39.1%)	Fracture *n =* 6 (26.1%)	Tooth fracture *n =* 4 (17.4%)	Dental concussion *n =* 1 (4.3%)
Upper Extremities	Fracture *n* = 7 (30.4%)	Contusion *n =* 5 (21.7%)		
Chest	Rib fracture *n* = 3 (13%)	Vertebral fracture *n =* 1 (4.3%)	Pneumothorax *n =* 1 (4.3%)	
Abdomen	Free intra-abdominal fluid *n* = 1 (4.3%)	Contusion *n =* 1 (4.3%)	Lumbago *n =* 1 (4.3%)	
Lower Extremities	Fracture *n* = 6 (26.1%)	CLW *n =* 2 (8.7%)	Ligament Injury *n =* 1 (4.3%)	Contusion *n =* 1 (4.3%)
Skin	Abrasion *n* = 13 (56.5%)			

Abbreviations: CCT = craniocerebral trauma, CLW = contused lacerated wound, ICH = intracranial haemorrhage.

## Data Availability

Not applicable.

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
