# Peer review of "Trauma Characteristics Associated with E-Scooter Accidents in Switzerland—A Case Series Study"

_ijerph, 2023, doi:10.3390/ijerph20054233_

Round 1

Reviewer 1 Report

This is a paper describing a small cohort of 23 patients sustaining injuries associated with e-scooter use. The novel aspect is that is from Switzerland, however no other novel factor as to why Switzerland would make a different study setting to other, similar, larger studies in the literature (Los Angeles, Tel Aviv, London, Liverpool). The study would benefit strongly from a comparative group, such as e-bikes or pedal bicycles.

The correlation with lack of helmet use needs to be better supported to give as a conclusion, otherwise, it would be correlation, not causation.

Abstract 

Line 21: Median needs Standard deviation

Line 22: ‘vaccidents’

Line 27: An overview of the surgeries would be helpful for the abstract 

Introduction

Table 1: For the patient group that had surgery, more details on the injury type and surgery received would be useful

Discussion: ‘alcoholised e-scooter drivers’ – suggest inebriated

Author Response

 Bern, 14.February 2023

Dear Editor-in-Chief and Reviewers,

On behalf of my co-authors, I would like to kindly thank you for your review of our manuscript, entitled “Trauma characteristics associated with e-scooter accidents in Switzerland - a case series study”. I would also like to thank you for giving us the opportunity to re-submit our article and for your very helpful comments.

As such, the authors have carefully assessed the Reviewer’s comments and the responses to the Reviewer’s concerns are as follows:

  1. I found the description of the time trend difficult to follow: 5 cases in 8 months, 10 in 12 months, and 9 in 10 months. Yes, also average case numbers per month do not fully suffice, because in the years 2019 and 2022 non-summer months were over-represented. I guess a formal Poisson regression per month controlling for season and year is not possible?

Author response: Thank you for this meaningful comment! We agreed that the years (2019-2022) are difficult to compare because summer month are over-represented in 2019 and 2022. We therefore refrain from presenting the time trend at all. Unfortunately, the small sample size does not allow for a meaningful Poisson regression per month controlling for season and year.

The first paragraph was adapted as follows:

“A total of twenty-three (n = 23) accidents with e-scooters was reported during the 30-month study period from May 2019 until October 2021. Less than half of all patients (39.1%, n = 9) were female and 60.9% male (n = 14). Patients’ ages ranged from 13 to 63 years, with a mean age of 36 (STD 14.8) years. Detailed patient characteristics are shown in Table 1.”

  1. The study would benefit strongly from a comparative group, such as e-bikes or pedal bicycles.

Author response: Thank you for this important comment. We discussed this point in our authors' group and decided to publish the results of this case series first in order to focus entirely on e-scooter accidents, as no Swiss data on this new group of micromobility devices has been published so far. Several studies on e-bike accidents are already available. However, a comparison with e-bikes, which are also a very popular means of transport in Switzerland, is certainly worth pursuing in the future.

  1. The correlation with lack of helmet use needs to be better supported to give as a conclusion, otherwise, it would be a correlation, not causation.

Author response: Thank you very much for your comment. Due to the imprecise data regarding helmet protection in the majority of the series, we have refrained from a statistical evaluation. However, we searched the literature for further studies on this topic and found two very recent additions to the literature that investigated the use of helmets in e-scooter accidents in simulations with human body models. As these results are of great interest for our conclusion, that wearing helmets seems to be beneficial in preventing severe head injuries, we decided to add a few sentences to our discussion:

“Two very recent studies evaluated the use of protective helmets in prevention of severe head injuries in e-scooter accidents, using simulations with human body models.[23, 24] The results suggest that helmets have the potential to reduce the impact forces experienced by the head during a fall and thus, lower the risk of head injuries. It's important to note that further research is necessary to determine the optimal design and effectiveness of helmets in reducing head injury risk in e-scooter accidents. However, based on the current evidence, both models suggest that wearing a helmet while riding an e-scooter is a simple and effective way to reduce the risk of head injuries in the event of a fall.”

  1. Abstract, Line 21: Median needs Standard deviation

Author response: Thank you for this comment. Age follows a normal distribution (Skewness 0.273, Kurtosis -0.756, Shapiro-Wilk test 0.450). We now consistently report the mean and standard deviation for age in the manuscript.

  • Mean 35.8 (STD 14.8) years

For clarification we added the following paragraph to the methods:

“The distribution of continuous variables were described as mean and standard deviation for normally distributed variables. Median and interquartile range was used to report not normally distributed variables. The distribution of categorical data was reported as numbers and percentages.”

  1. Line 22: “vaccidents”

Author response: Please excuse this mistake, it was corrected.

  1. Line 27: An overview of the surgeries would be helpful for the abstract. /
    Table 1: For the patient group that had surgery, more details on the injury type and surgery received would be useful.

Author response: Thank you for these two comments, that we decided to summarize in one answer. We entered patient records again and searched for the surgeries, that were performed on these patients in the first few days of their hospitalization after the e-scooter accident. Two of the patients were heavily injured and had to undergo many more procedures in the following. We decided to focus on the surgeries that were performed in the very beginning and tried to specify on the one hand the responsible specialty and on the other hand the procedures that were performed. Please consider that Table 1 has very narrow columns and therefore, the details that were inserted under the headline “Outcome” might be difficult to read in the current formatting.

  1. Discussion: “alcoholised e-scooter drivers” – suggest inebriated

Author response: Thank you very much for this suggestion. The vocabulary was adapted.

  1. The discussion is slightly disjointed, where alcohol use is brought up in the second paragraph and then again in the third paragraph after the incidence in the evening. I think it would be a more cogent if alcohol was discussed completely in the second paragraph and discussion of time of day is move prior to discussion of seasonality.

Author response: Thank you very much for this helpful comment. We very much agree with you on this point and have moved the passage

“A significant number of e-scooter accidents in Switzerland occurred under the influence of alcohol. Alcohol use is of concern because it impairs driver’s judgment and thus may lead to accidents. Prevention campaigns raising aware-ness of the risks of driving e-scooters under the influence of alcohol should be specifically promoted. In addition, more rigorous alcohol controls and consecutive punishment of e-scooter drivers under the influence of alcohol could help prevent future accidents.”

from the third paragraph to the second. In this way we achieve a coherent discussion of the role of alcohol consumption in e-scooter accidents.

In addition, we would like to propose to combine the fourth paragraph with the third. The fourth paragraph discusses possible measures to reduce the high frequency of accidents at night and thus ideally concludes the third paragraph.

  1. pg.8 ln 273: Does the 26.1% of injuries caused by other individual apply to all of the automobile-scooter traumas? Data from other studies show that automobile and scooter collisions account for more severe traumas.

Author response: Thank you for your question. Yes, the 26.1% of injuries caused by other individuals refers to automobile-scooter collisions in the study. Out of 6 patients, 4 had direct collisions with cars, with 2 of them being more severely injured, further emphasizing your point concerning the severity of traumas. 1 patient collided with a parked car. The remaining 2 patients were injured during evasive maneuvers.

We included this further information in results:

“Four patients had direct collisions with cars, one of them with a parked car. The remaining two patients were injured during evasive maneuvers.”

  1. Can you give us a number of how many patients were not included in the study? So that the reader will have an insight on the overall number? I am asking because n=23 is a very low number for that long period of analysis.

Author response: Thank you for your interest. No patients were excluded from the study. To clarify this matter a sentence was added to the results:

All patients agreed to the inclusion of their anonymized data in the research.

We are aware that the number of patients included is small. However, we would like to point out that the e-scooter rental services of Tier and Voi were only introduced in Bern during the study period, in January 2021.

  1. There is one patient who was wearing a helmet, was that a private e-scooter user? If yes, you could you name that in the text.

Author response: Thank you very much for this question. We have reexamined the medical records but have not found any information on the ownership of the e-scooter involved in the accident. Therefore, we unfortunately cannot include this information in the manuscript.

  1. As a lot of patients in your study suffer head and face injuries and the total number is with 23 very low for such a study I would recommend to add more literature in the discussion so that the article gains a "review" character.

Author response: Thank you for your comment, which motivated us to search the literature again. In addition to your valuable recommendations, we found some other recently published studies. The work by Grill et al., as well as two very interesting studies by Gan-El et al. and Mair et al. gave us complementary information that we could use to emphasise our arguments. Additionally, during the literature search we discovered two interesting simulation studies on the use of helmets by e-scooter users. All new references were included in the discussion at various points to give our article more of a review character, and added to the bibliography:

Gan-El et al. (2022) reported intoxication in 30% of their patients, Blomberg et al. (2019) in 36.6%, Mair et al. (2021) in 36.7%, and Grill et al. (2022) in 52.5% of the e-scooter users admitted to hospital.[5, 25-27]”

“Considering the time of day, authors from other studies describe too an increased number of e-scooter accidents during evening and night-time hours, which is in line with our findings.[5,8,9, 25-27]”

“Mair et al. (2021) calculated for night-hours from 10pm-6am that 72.7% of all patients admitted during this time span had consumed alcohol, which is almost the double of their findings concerning the over-all rate of inebriated e-scooter drivers.[26]”

“Gan-El et al. (2022), as well as Moftakhar et al. (2021), both reported too most injuries in the upper limbs, closely followed by the head and neck area.[9, 27] The frequent occurrences of head and neck injuries in conjunction with injuries to the upper extremities might indicate that drivers attempted to break their fall before hitting the ground.»

  1. Please overwork your conclusions as this part is mostly read by the city governments and will have an influence on the argumentation when it comes to new traffic safety measures. The responsible parties could then argue that even the research group from Bern are recommending.... Please be aware of your responsibility.

Author response: Thank you for this thought-provoking comment. We are very aware that such statements have consequences, which is why we are very careful with the wording. However, we would still like to make a comment on helmet protection, as this helps to underline the review character of the study. We have therefore weakened the conclusion:

“As literature available underlines the positive effects of helmet protection in cyclists as well as e-scooter pilots, it is beneficial for both groups to use them.”

We have also searched the literature again and found two brand-new studies on the subject of helmets and e-scooter injuries. Based on several simulations, both studies recommend wearing a helmet when riding an e-scooter. Furthermore, we described the existing foldable helmets from Tier and made a suggestion, how their usage could be increased.

“Two very recent studies evaluated the use of protective helmets in prevention of severe head injuries in e-scooter accidents, using simulations with human body models.[23, 24]. The results suggest that helmets have the potential to reduce the impact forces experienced by the head during a fall and thus, lower the risk of head injuries. It's important to note that further research is necessary to determine the optimal design and effectiveness of helmets in reducing head injury risk in e-scooter accidents. However, based on the current evidence, both models suggest that wearing a helmet while riding an e-scooter is a simple and effective way to reduce the risk of head injuries in the event of a fall. Each country must consider whether its population would benefit from an official helmet obligation with possible fines for violations. Based on the available data, however, there seems to be a general benefit of helmet protection. The rental company Tier offers foldable helmets for free.[29] On their “How to use” webpage, however, they do not include any advice or information about the usage of the in the e-scooters integrated helmets. Only in a youtoube-video this step is further explained. In our opinion, to increase the usage of helmets amongst e-scooter drivers, the availability of the foldable helmets should be promoted more strongly.”

In addition, we have added to the paragraph on drunk e-scooter riders, in particular further references to underline the danger of alcohol concerning the evolution of e-scooter accidents, as well as Voi's in-app reaction test:

Gan-El et al. (2022) reported intoxication in 30% of their patients, Blomberg et al. (2019) in 36.6%, Mair et al. (2021) in 36.7%, and Grill et al. (2022) in 52.5% of the e-scooter users admitted to hospital.[5, 25-27] This is similar to our findings of 43.5%. Oksanen et al. (2020) even found as many as 91% of all patients intoxicated, with an average blood alcohol level of 1.7‰.[13] In September 2020, one of the rental companies, Voi, has introduced an in-app reaction test to investigate users’ fitness to drive.[28] However, riders are still able to unlock the scooter - regardless of whether or not they pass this test.

I would also like to thank the editors and reviewers for their careful and conscientious review. We have highlighted all changed passages in the manuscript in tracking mode. If you need any further information, please do not hesitate to contact me.

Yours sincerely,

Ava Bracher, on behalf of all authors

Reviewer 2 Report

The introduction is well-written and concise, demonstrating the need for such studies on the effects of e-scooters accidents in urban environments. The stastitical methods are adequate for the comparisons being made, and the results are clear and concise. 

The discussion is slightly disjointed, where alcohol use is brought up in the second paragraph and then again in the third paragraph after the incidence in the evening. I think it would be a more cogent if alcohol was discussed completely in the second paragraph and discussion of time of day is move prior to discussion of seasonality. 

pg.8 ln 273: Does the 26.1% of injuries caused by other individual apply to all of the automobile-scooter traumas? Data from other studies show that automobile and scooter collisions account for more severe traumas. 

Author Response

(The authors gave the same response as above.)

Reviewer 3 Report

Dear Authors,

thank you for submitting your work. This is a hot topic and cities all over the world are facing the same problem and are looking for measures to get this accidents under "control". It was a pleasure to read and review this manuskript.

Nevertheless there are some thiong to add and to work on.

1) can you give us a number of how many patients were not included in the study? So that the reader will have a insight on the overall number? 

I am asking because n=23 is a very low number for that long period of analysis.

2) There is one patient who was wearing a helment, was that a private e-scooter user? If yes you could you name that in the text.

3) As a lot of patients in your study suffer head and face injuries and the total number is with 23 very low for such a study i would recomend to ad more literature in the discussion so that the article gains a "review" character.

https://www.sciencedirect.com/science/article/abs/pii/S101051822200066X?via%3Dihub

https://link.springer.com/article/10.1007/s00402-020-03589-y

All 3 Papers are representing german speaking cities and are located in the middle of europe. So that a similarity could be found. This would give your work as I mentioned a clear more review character of the situation in europe.

4) Please overwork your conclusions as this part is mostly read by the city gouverments and will have an influance on the argumentation when it comes to new traffic safety measures. The responsible parties could then argue that even the research group from Bern are recommending.... Please be aware of your resposibilty

The e-scooter companies are able to track all rides. After some clear data showed the influance of alcohol on the injuries some e-scooter companies introduced a pre-ride reaction test to get rid of the drunken users.

Further in Australia a helmet is mandatory and is placed on every e-scooter that is for rent.

Basically I really like for manuscript biut please be aware of the political impact you have. As I said 23 is a very low number of patients treated so you have to do more in the disscussion part. 

Best regards

Author Response

(The authors gave the same response as above.)
